# Nab-Paclitaxel and Gemcitabine as First-Line Treatment of Metastatic Ampullary Adenocarcinoma with a Novel *R-Spondin2* RNA Fusion and *NTRK3* Mutation

**DOI:** 10.3390/biomedicines11082326

**Published:** 2023-08-21

**Authors:** Maryknoll P. Linscott, Havell Markus, Mackenzie Sennett, Catherine Abendroth, Nelson S. Yee

**Affiliations:** 1Medical Scientist Training Program, Pennsylvania State University College of Medicine, Hershey, PA 17033, USA; mpalisoc@pennstatehealth.psu.edu (M.P.L.); hmarkus@pennstatehealth.psu.edu (H.M.); msennett@pennstatehealth.psu.edu (M.S.); 2Department of Pathology, Penn State Health Milton S. Hershey Medical Center, Hershey, PA 17033, USA; cabendroth@pennstatehealth.psu.edu; 3Division of Hematology-Oncology, Department of Medicine, Pennsylvania State University College of Medicine, Hershey, PA 17033, USA; 4Next-Generation Therapies Program, Penn State Cancer Institute, Penn State Health Milton S. Hershey Medical Center, Hershey, PA 17033, USA

**Keywords:** ampullary adenocarcinoma, pancreaticobiliary, nab-paclitaxel, gemcitabine, molecular profiling, circulating tumor DNA, *R-Spondin2*, *NTRK3*

## Abstract

Ampullary adenocarcinoma is a rare malignancy that lacks standard systemic treatment. We describe a case of recurrent metastatic ampullary adenocarcinoma of the pancreaticobiliary subtype treated with nanoparticle albumin-bound (nab)-paclitaxel and gemcitabine as first-line treatment. This report also highlights the molecular profile of the ampullary adenocarcinoma and circulating tumor DNA (ctDNA). This is a case of pancreaticobiliary ampullary adenocarcinoma in a 67-year-old woman who initially presented with painless jaundice. Endoscopic and imaging evaluation revealed biliary ductal dilation secondary to an ampullary mass. Pathology confirmed the diagnosis of ampullary adenocarcinoma of the pancreaticobiliary subtype. She underwent surgical resection of the tumor, followed by adjuvant chemotherapy with gemcitabine and capecitabine. The tumor subsequently recurred in the liver. She received palliative chemotherapy with nab-paclitaxel and gemcitabine, resulting in an objective tumor response for 14 months. Molecular profiling of the tumor and ctDNA revealed a novel *MATN2-RSPO* RNA fusion and a novel *NTRK3* mutation, respectively. Our report suggests that long-term durable response can be achieved in metastatic pancreaticobiliary ampullary adenocarcinoma using nab-paclitaxel and gemcitabine. Molecular profiling of the tumor identified a novel *R-Spondin2* RNA fusion and *NTRK3* mutation that can be potentially targeted for treatment.

## 1. Introduction

Adenocarcinoma of the ampulla of Vater, or ampullary adenocarcinoma (AAC), is a rare malignancy that comprises 0.2% of gastrointestinal cancers [1,2,3]. AAC develops from a mucosal landmark where the common bile duct and pancreatic duct converge. Depending on the epithelium of origin, AACs can be classified into two main histologic subtypes: intestinal and pancreaticobiliary [4,5]. Although not typical, a mixed histologic subtype also carries ambiguous features from both subtypes [6].

A growing number of studies demonstrate that the pancreaticobiliary subtype correlates with worse patient outcomes than the intestinal subtype [7,8,9,10]. Although many studies have shown that patients with AAC have improved survival compared to other peri-ampullary and hepato-pancreato-biliary malignancies [10,11], this observation could be attributed to early detection of the disease. Patients with AAC usually present with obstructive jaundice, whereas those with other cancers typically remain asymptomatic until tumor progression.

Curative surgery, usually pancreaticoduodenectomy (Whipple procedure), is possible in at least 50% of patients with AAC as compared to fewer than 10% in those with pancreatic adenocarcinoma [12]. Nonetheless, up to 50% of AAC cases ultimately recur [2,13], justifying the need for systemic adjuvant chemotherapy. Given its rarity, randomized clinical trials focused on AAC are scarce. The external validity of available clinical studies on AAC cases also remains limited. These studies analyzed AAC cases as a minor component in a combined group, which commonly includes pancreatic, small bowel, and other biliary malignancies [14,15,16]. Due to the lack of clinical trials that focus on AAC, there is no consensus on the optimal management of this type of tumor [17].

Without standard treatment options in rare malignancies, and even more so in advanced cancers, an empirical approach weighing the risks and benefits of off-label use of chemotherapies is used [18]. While gemcitabine-based treatments are considered conventional regimens for advanced AAC and other biliary tract cancers, AACs have also been treated with chemotherapies for pancreatic or small bowel adenocarcinomas [14,19,20,21,22]. Actionable genetic alterations, including mismatch repair deficiency, microsatellite instability-high (MSI-H), and point mutations, could also be targeted using tissue-agnostic immunotherapies and targeted agents, especially in advanced disease [23,24]. With the advent of these therapies, tumor tissue profiling platforms have been developed to provide biomarker information and guide treatment decisions.

In this report, a case of a patient with metastatic AAC of the pancreaticobiliary subtype that responded to gemcitabine/nab-paclitaxel for 14 months is presented. The molecular profiles of the patient’s primary tumor as well as the biopsied blood specimens during treatment with gemcitabine and nab-paclitaxel are described.

## 2. Case Description

The patient is a 67-year-old Caucasian woman who initially presented with painless jaundice. Past medical history included hypothyroidism and chronic kidney disease. Family history was significant for father with prostate cancer, mother with colon cancer, and brother with laryngeal cancer. She denied any history of smoking and alcohol or substance abuse. She had not travelled outside the country within the past 30 days. Medications included levothyroxine, multivitamins, calcium carbonate/vitamin D, and ascorbate calcium.

Diagnostic evaluation with laboratory tests and chest/abdominal/pelvic computed tomography (CT) with contrast was performed. Laboratory results were notable for elevated total bilirubin (14.6 mg/dL; normal 0–1.2 mg/dL), direct bilirubin (10.9 mg/dL; normal 0–0.3 mg/dL), alanine transaminase (694 unit/L; normal 0–33 unit/L), aspartate transaminase (393 unit/L; 0–32 unit/L), carbohydrate antigen 19-9 (CA 19-9; 245.3 unit/mL; normal <36 unit/mL), and carcinoembryonic antigen (CEA; 5.3 ng/mL; normal <4.8 ng/mL). Complete blood count (CBC) with differential, comprehensive metabolic panel (CMP), iron, hemoglobin, prothrombin time, and thyroid-stimulating hormone (TSH) were within normal limits. Tests for hepatitis A virus, hepatitis B virus, hepatitis C virus, and cytomegalovirus were negative. Past infection with Epstein–Barr virus was detected.

CT scans revealed an ampullary mass and biliary ductal dilation involving the pancreatic, intrahepatic, and extrahepatic bile ducts, which extended to the ampulla (Figure 1). No tumor metastasis was detected.

Endoscopic retrograde cholangiopancreatography (ERCP) was conducted with sphincterotomy and placement of stents in the bile duct and pancreatic duct. An ampulla mass was detected on endoscopic ultrasonograph (EUS). Histological examination of the biopsied ampulla mass confirmed AAC (Figure 2A). The patient then underwent pancreaticoduodenectomy (Whipple procedure).

The resected tumor revealed intra-ampullary and peri-ampullary adenocarcinoma (Figure 2B), pancreaticobiliary type (Figure 2C), pT2 and pN1, with histologic grade 2. The tumor measured 1.2 cm (greatest dimension) and extended beyond the sphincter of Oddi and into the muscularis propria of the duodenum. One of the 40 regional lymph nodes resected was involved in the tumor. The resection margins (pancreatic parenchyma, bile duct, proximal, and distal margins) were uninvolved in the tumor.

The patient received adjuvant chemotherapy (May 2020) with gemcitabine (800 mg/m^2^ IV on day 1, day 8, and day 15) and capecitabine (500 mg ii PO every 12 h on day 1 to day 21) of every 28-day cycle. Due to neutropenia, the dosage of gemcitabine in cycle 3 was decreased to 700 mg/m^2^ (July 2020). Peg-filgrastim was also administered. Follow-up CT scans (July 2020) revealed a new hypodense lesion in segment 7 of the liver (Figure 3A), also shown as a rim-enhancing 1.1 cm lesion through magnetic resonance imaging (MRI). The hepatic lesion was biopsied, and pathological examination confirmed metastatic adenocarcinoma (Figure 3B). Due to recurrent tumor metastasis, the treatment with gemcitabine and capecitabine was discontinued (July 2020).

The patient was then started on a combination chemotherapy conventionally used in metastatic pancreatic adenocarcinoma—nab-paclitaxel and gemcitabine (July 2020). Nab-paclitaxel is a microtubule inhibitor that promotes the assembly of microtubules from tubulin dimers and stabilizes microtubules by preventing depolymerization, resulting in inhibition of the normal dynamic reorganization of the microtubule network during mitosis. Gemcitabine exerts cytotoxic effects by inhibiting DNA synthesis by blocking ribonucleotide reductase and competing with dCTP for incorporation into DNA. Considering her history of neutropenia, both chemotherapeutic drugs were initially infused at 60% full dose: nab-paclitaxel 75 mg/m^2^ and gemcitabine 600 mg/m^2^ on day 1, day 8, and day 15 of every 28-day cycle). Due to persistent neutropenia, the dosage of gemcitabine was further reduced to 500 mg/m^2^ prior to starting cycle 3. Having completed six cycles of nab-paclitaxel and gemcitabine, the patient demonstrated continued tumor response, as shown by the decrease in size of the hepatic lesion and the tumor marker CA 19-9 trending down (Figure 4). The patient tolerated nab-paclitaxel and gemcitabine reasonably well, except for transient nausea and mild paresthesia in the toes.

Molecular profiling (by Caris^®^ Molecular Intelligence, Caris^®^ Life Sciences, Phoenix, AR, USA) was conducted on the surgically resected AAC to explore treatment options (Table 1). The results indicated mutations in *CDKN2A* DNA (pathogenic variant, exon 2, codon 172C>T, variant frequency 51%, protein R58*) and *KRAS* DNA (pathogenic variant, exon 2, codon 35G>A, variant frequency 45%, protein G12D). Alteration in *RSPO2* RNA showed a likely pathogenic fusion (exon 3, protein MATN2-RSPO2), whereas fusion was not detected in *NTRK1/2/3*. A variant of uncertain significance was detected in *ATM* DNA (exon 8, codon 1009C>T, variant frequency 45%, protein R337C). The tumor was also negative for *ERBB2* amplification, PD-L1 expression, and PTEN deletion. Microsatellite instability was low, and mismatch repair status was proficient. The tumor mutational burden was low (1 mutation/Mb).

The patient continued to receive three more cycles of chemotherapy, with both drugs’ dosages reduced to 50% of the full dose due to persistent neutropenia (January 2021). Follow-up CT scans (April 2021) showed a stable lesion in segment 7 of the liver (Figure 5A). In addition to radiological studies, blood specimens were collected to analyze circulating tumor DNA (ctDNA, by Guardant360^®^, Guardant^TM^, Redwood City, CA, USA) in order to help monitor tumor response to chemotherapy. CtDNA and MSI-high were undetected after cycle 9 of nab-paclitaxel and gemcitabine (April 2021). However, prior to cycle 12 (4 June 2021), a ctDNA mutation was identified in *NTRK3* (R645H, 0.1%). A slight increase in the conspicuity of the segment 7 lesion on CT scans was observed (July 2021). Subsequently, cycle 13 treatment was administered with the dosage of nab-paclitaxel increased to 70% full dose.

CT scans (August 2021) suggested that the patient’s tumor had progressed, as indicated by further enlargement of the hepatic lesion (18 mm, previously 9 mm; Figure 5B) and continuous rising of serum CA 19-9 (Figure 4). Treatment with nab-paclitaxel and gemcitabine was discontinued. She was considered for any eligible clinical trial (Clinicaltrials.gov Identifier: NCT04003896, NCT02521844, NCT04484142) based on genetic alterations in *KRAS* and *CDKN2A*, *RSPO2*, and *NTRK3*, respectively (Table 1). Palliative microwave ablation or Y-90 radioembolization of the metastatic tumor in the liver was also considered.

## 3. Discussion

In this case report, we present a patient who received nab-paclitaxel and gemcitabine as palliative systemic treatment for recurrent metastatic AAC of the pancreaticobiliary subtype. Moreover, we provide the data of multi-platform molecular profiling for identifying actionable biomarkers as well as monitoring tumor response to treatment. To our knowledge, this is the first reported case of using nab-paclitaxel and gemcitabine as a first-line systemic treatment for metastatic AAC with durable response (14 months).

There are a few previously reported cases in which patients with metastatic AAC were initially treated with fluorouracil-based chemotherapy prior to nab-paclitaxel and gemcitabine. In a patient with recurrent metastatic AAC with a mixed subtype that had progressed following treatment with FOLFOX and then FOLFIRI, a tumor response of 9 months using nab-paclitaxel and gemcitabine was demonstrated [28]. In another patient with pancreaticobiliary-type AAC that had progressed following treatment with FOLFOX for 10 months and then one dose of FOLFIRI, treatment with nab-paclitaxel, gemcitabine, and cisplatin produced a tumor response of at least 12 months [29]. In a patient with metastatic pancreaticobiliary-type AAC that had progressed following treatment with FOLFOX for 5 months and then FOLFIRINOX for 2 months and palliative radiation therapy, treatment with nab-paclitaxel and gemcitabine produced a tumor response of at least 10 months [29]. Those reported patients who received nab-paclitaxel and gemcitabine with or without cisplatin following prior treatment with 5-fluorouracil-based chemotherapy lend support to our case.

As a standard practice in the management of advanced or metastatic carcinoma, including AAC, molecular profiling of the tumor and analysis of ctDNA are routinely conducted and typically covered by health insurance in order to explore actionable biomarkers to help select treatment using molecularly targeted agents. In this report, tumor molecular profiling by multi-platform analyses revealed typical as well as novel genetic alterations (Table 1). DNA sequencing of the surgically resected AAC revealed genetic alterations in *KRAS* and *CDKN2A*, which are common pathogenic variants in AAC, as previously reported [25,26,27]. Cyclin-dependent kinases 4 and 6 (CDK4/6) act downstream of KRAS and CDKN2A, and they are potential targets of anti-cancer therapy. A clinical trial to investigate abemaciclib (an inhibitor of CDK4/6) in advanced or metastatic biliary tract carcinoma, including AAC (Clinicaltrials.gov Identifier: NCT04003896), can be considered for this patient [30].

Moreover, a likely pathogenic RNA fusion was identified in the *RSPO2* gene, which encodes a member of the R-spondin family of secreted proteins that can potentiate canonical Wnt signaling [31]. Although this specific fusion has not been previously identified, it is considered potentially oncogenic, as the exon 3 of *RSPO2* is joined in-frame with the exon 4 of *MATN2,* which provides a signal sequence. Chromosomal translocations that result in fusion of *RSPO2* with genes such as *EIF3E*, *EMC2*, *PVT1*, and *HNF4A* have been identified in a variety of human cancers [32,33,34]. A preliminary study has shown an association between *RSPO* fusions and the co-activation of the MAPK pathway and modulation of PD-L1 expression in colorectal carcinoma [35]. These data suggest malignant tumors, including AAC, that harbor RSPO2 fusion can be potentially treated with Wnt-targeted agents and anti-PD-1/PD-L1 antibodies. A clinical study to investigate ETC-1922159 (a porcupine inhibitor that blocks post-translational palmitoylation of Wnt ligands and inhibits their secretion, preventing the activation of Wnt ligands, interfering with Wnt-mediated signaling, and inhibiting cell growth in Wnt-driven tumors) either as a single agent or in combination with pembrolizumab (anti-PD-1 antibodies) in advanced solid tumors is recruiting (Clinicaltrials.gov Identifier: NCT02521844) [36].

We utilized a commercial liquid biopsy sequencing platform (Guardant 360^®^) to monitor treatment response and identify actionable resistance mechanisms. Although no *NTRK 1/2/3* fusion was initially detected in the primary tumor, a *NTRK3* (R645H) mutation was detected in the ctDNA. Of note, a DNA alteration in the primary tumor was detected in *NTRK1* (Table 1). This *NTRK1* variant is considered likely benign since it has been identified in the germline of several individuals and has not been found as a somatic mutation in cancers. However, the genetic alteration in *NTRK3* is novel and of unknown clinical and functional significance. Previous studies have proposed the clinical relevance of *NTRK3* mutations in lung adenocarcinoma. A clinical study to investigate datopotamab deruxtican (a TROP2-directed antibody-drug conjugate) in advanced or metastatic non-small cell lung cancer with actionable genomic alterations, including *NTRK 1/2/3* mutation, is recruiting (Clinicaltrials.gov Identifier: NCT04484142) [37]. Moreover, various mutations in *NRTK3* were associated with a greater tumor burden and improved treatment outcomes using immune-checkpoint inhibitors compared to those without *NTRK3* alteration [38]. Whether our patient may benefit from immune-checkpoint inhibitors remains to be explored.

AACs arise within the peri-ampullary region where the distal bile duct, pancreatic duct, and duodenum merge, and they are associated with variable morphological subtypes and clinical phenotypes. The genomic profile of this reported patient, along with the previously published genetic alterations of AAC, is in agreement with the known molecular heterogeneity of this disease [25,26,27]. This may be attributed to the multiple cells-of-origin of AAC, inter-patient variations, and any additional acquired mutations during tumorigenesis. Using genomic-classifier methodology provided insight into the heterogeneity of AAC by correlating the histological-genomic subtypes with various genetic mutations, and suggested risk stratification based on the identification of distinct clinical subtypes may help guide the selection of personalized therapies [39].

## 4. Conclusions

Standard guidelines for the optimal management of advanced or metastatic AAC remain elusive due to the rarity of this malignancy. This study describes a case of metastatic AAC of the pancreaticobiliary subtype that recurred during adjuvant treatment with gemcitabine and capecitabine. Remarkably, a durable response (14 months) was achieved using nab-paclitaxel and gemcitabine, one of the standard palliative treatments for advanced pancreatic adenocarcinoma. Tumor molecular profiling revealed *RSPO2, KRAS,* and *CDKN2A* alterations prior to nab-paclitaxel and gemcitabine treatment, whereas a *NTRK3* mutation was associated with the recurrent tumor during treatment. This case report supports the utilization of nab-paclitaxel and gemcitabine in treating advanced AAC. Future clinical trials are indicated to investigate this combination as a first-line treatment for advanced or metastatic AAC. Moreover, this report also highlights the utility of molecular profiling platforms in guiding treatment decisions for patients with rare and aggressive diseases, such as AAC.

## Figures and Tables

**Figure 1 biomedicines-11-02326-f001:**
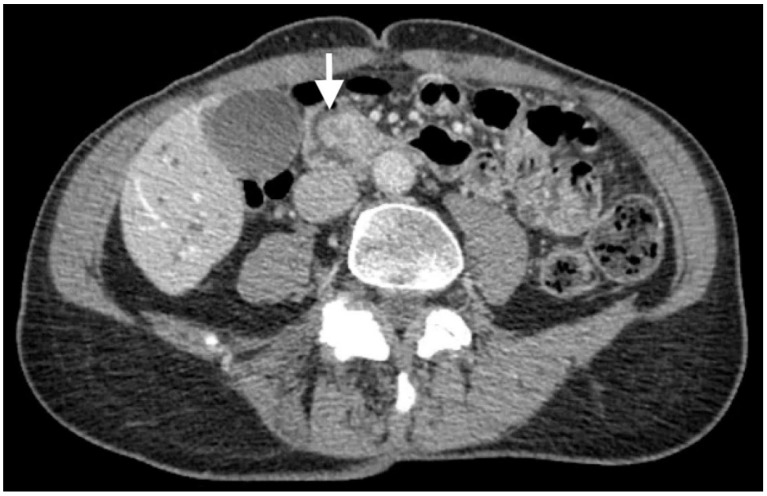
CT scans depicting an ampullary mass (white arrow) and biliary ductal dilation.

**Figure 2 biomedicines-11-02326-f002:**
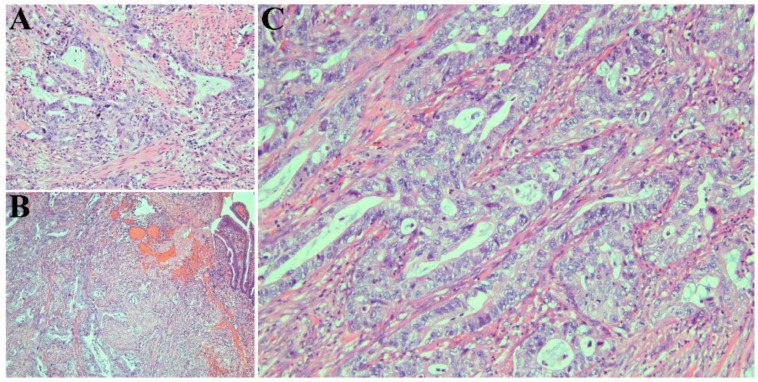
(**A**) Ampullary biopsy demonstrating adenocarcinoma with pancreaticobiliary morphology. Cells have round vesicular nuclei and polygonal cytoplasm and form rounded glands (hematoxylin and eosin, ×500). (**B**) Resected tumor confirming ampullary adenocarcinoma (hematoxylin and eosin, ×125). (**C**) Higher magnification view of resected tissue depicting pancreaticobiliary morphology (hematoxylin and eosin, ×500).

**Figure 3 biomedicines-11-02326-f003:**
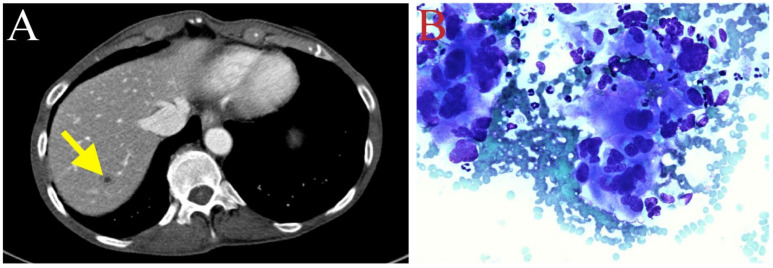
(**A**) New hypodense lesion in hepatic segment 7, suggesting tumor recurrence in the liver (depicted with yellow arrow). (**B**) Fine-needle aspiration biopsy of liver metastasis. Three-dimensional glandular aggregates of neoplastic cells possessed of moderate volume of finely vacuolated cytoplasm demonstrating marked nuclear pleomorphism (Diff Quick, ×500).

**Figure 4 biomedicines-11-02326-f004:**
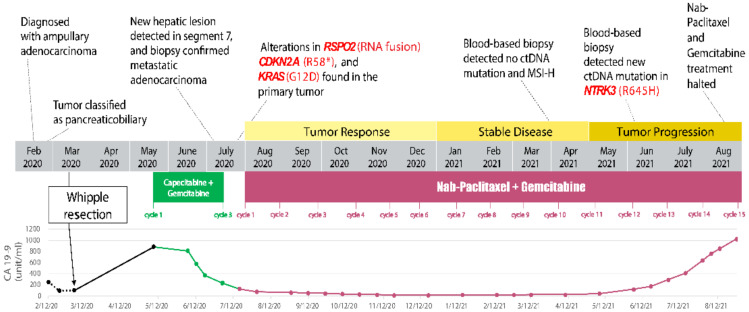
Timeline of significant events in the patient’s disease course. Changes in treatment, CA 19-9 level, molecular profiling result, and disease outcome are summarized in this schematic. ^#^ Indeterminate amino acid.

**Figure 5 biomedicines-11-02326-f005:**
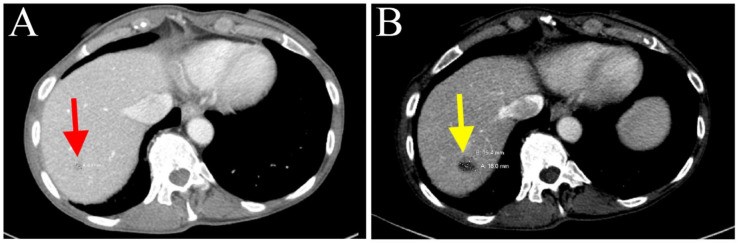
(**A**) Stable lesion in hepatic segment 7 after cycle 9 of gemcitabine/nab-paclitaxel treatment (depicted with red arrow). (**B**) Tumor enlargement after cycle 13 (depicted with yellow arrow).

**Table 1 biomedicines-11-02326-t001:** Molecular profile of this reported patient’s tumor compared to the genetic alterations in ampullary adenocarcinoma with pancreaticobiliary subtype previously published in the literature.

This Case Report (*n* = 1)	Yachida et al., 2016 [25] (*n* = 66)	Gingras et al., 2016 [26] (*n* = 71)	Ferchichi et al., 2018 [27] (*n* = 15)
*KRAS* exon 2, c.35G>A, p.G12D	*KRAS* (68%)	*KRAS* (72%)	*KRAS* p.G12D (27%) *KRAS* p.G12A (13%) *NRAS* p.G12D (7%)
*CDKN2A* exon 2, c.172C>T, p.R58 *	*CDKN2A* (9%)	*CDKN2A* (16%)	NR
*RSPO2* (RNA fusion) *MATN2* exon 4: *RSPO2* exon 3 *MATN2*-*RSPO2*	NR	NR	NR
*ATM* c.1009C>T p.R337C	NR	*ATM* (10%)	NR
*NTRK1* Exon 6, c.640C>T, p.R214W	NR	NR	NR
NR	*TP53* (67%)	*TP53* (72%)	NR
NR	*SMAD4* (20%)	*SMAD4* (18%)	NR
NR	*CTNNB1* (15%)	NR	NR
NR	*ERBB3* (14%)	NR	NR
NR	*GNAS* (12%)	NR	NR
NR	*CDH10* (12%)	NR	NR
NR	*ELF3* (11%)	NR	NR
NR	NR	*FBXW7* (8%)	NR
NR	NR	*PIK3CA* (13%)	NR
NR	NR	*ARID1A* (13%)	NR
NR	NR	*APC* (11%)	NR
NR	NR	*TGFBR2* (10%)	NR
NR	NR	*FBXW7* (8%)	NR

*MATN2-RSPO2* RNA fusion was detected in which the exon 3 of *RSPO2* is joined in-frame with the exon 4 of *MATN2*. *ATM* c.1009C>T is a variant of uncertain significance since it has been reported in both the population database and as a somatic mutation in cancer. *NTRK1* c.640C>T is likely a benign variant since it has been identified in the germline of several individuals and has not been found as a somatic mutation in cancers. c: codon; p: protein; NR: not reported. * Indeterminate amino acid.

## Data Availability

Deidentified raw data supporting the conclusions of this article will be made available by the authors upon request.

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
