# Peer review of "Nab-Paclitaxel and Gemcitabine as First-Line Treatment of Metastatic Ampullary Adenocarcinoma with a Novel R-Spondin2 RNA Fusion and NTRK3 Mutation"

_biomedicines, 2023, doi:10.3390/biomedicines11082326_

Round 1

Reviewer 1 Report

I am really grateful to review this manuscript. In my opinion, this manuscript can be published once some revision is done successfully. I made one suggestion and I would like to ask your kind understanding. This study presented a case for nab-paclitaxel and gemcitabine as the first-line treatment of metastatic ampullary adenocarcinoma with a novel R-Spondin2 RNA fusion and NTRK3 mutation. This study registered a timeline of the patient’s disease course including tumor classification, treatment, response and progression. This study compared the molecular profile of the patient’s tumor to those of existing literature as well. As the authors pointed out, however, there would exist a large degree of heterogeneity among ampullary carcinomas. I would like to ask the authors to elaborate this issue in the end of the manuscript.

Author Response

The authors would like to thank Reviewer 1's review and comments.

Authors' response:

The following paragraph has been added to Page 7, Discussion and highlighted in yellow color:

     AACs arise within the peri-ampullary region where the distal bile duct, pancreatic duct, and duodenum merge, and they are associated with variable morphological subtypes and clinical phenotypes. The genomic profile of this reported patient along with the previously published genetic alterations of AAC is in agreement with the known molecular heterogeneity of this disease [27-29]. This may be attributed to the multiple cells-of-origin of AAC, inter-patient variations, and any additional acquired mutations during tumorigenesis. Using genomic-classifier methodology, a recent report provided insight into the heterogeneity of AAC by correlating the histological-genomic subtypes with various genetic mutations, and suggested risk stratification based on the identification of distinct clinical subtypes may help guide the selection of personalized therapies [39].

Reviewer 2 Report

Dear Editor

This MS is described a case report of a patient with metastatic AAC of the pancreaticobiliary sub- 64 type, which responded to gemcitabine/nab-paclitaxel for 14 months. A complete study has been done. It is better to mention the mechanism of these drugs in the introduction or discussion.

Best Regards

Author Response

The authors would like to thank Reviewer 2's review and comments.

Authors' response:

The following sentences have been added to Page 4, and highlighted in yellow color:

… The patient was then started on a combination chemotherapy conventionally used in metastatic pancreatic adenocarcinoma—nab-paclitaxel and gemcitabine (July 2020). Nab-paclitaxel is a microtubule inhibitor that promotes the assembly of microtubules from tubulin dimers and stabilizes microtubules by preventing depolymerization, resulting in inhibition of the normal dynamic reorganization of the microtubule network during mitosis. Gemcitabine exerts cytotoxic effect by inhibiting DNA synthesis through blocking ribonucleotide reductase and competing with dCTP for incorporation into DNA.

Reviewer 3 Report

The manuscript entitled Nab-paclitaxel and gemcitabine as first-line treatment of meta- static ampullary adenocarcinoma with a novel R-Spondin2 RNA fusion and NTRK3 mutation. Overall, the manuscript is organized and detailed, and the writing is readable. It is a case report of 67 years old woman ampullary adenocarcinoma. The  report suggests that long-term durable response can be achieved in metastatic pancre-aticobiliary ampullary adenocarcinoma using nab-paclitaxel and gemcitabine.

the manuscript is very well written, since it is a case report of one patient. so there may be some variations when the sample size will be increased.

i have no further comments or suggestion for this particular manuscript. it can be accepted for publication.

Author Response

The authors would like to thank Reviewer 3's review and comments.